# Rac GTPase Signaling in Immune-Mediated Mechanisms of Atherosclerosis

**DOI:** 10.3390/cells10112808

**Published:** 2021-10-20

**Authors:** Cadence F. Lee, Rachel E. Carley, Celia A. Butler, Alan R. Morrison

**Affiliations:** 1Ocean State Research Institute, Inc., Providence VA Medical Center, Research (151), 830 Chalkstone Avenue, Providence, RI 02908, USA; cadence_lee@alumni.brown.edu (C.F.L.); rachel_carley@brown.edu (R.E.C.); celia.butler@va.gov (C.A.B.); 2Alpert Medical School, Brown University, Providence, RI 02912, USA

**Keywords:** Racs, atherosclerosis, statins, macrophage, inflammation, calcification

## Abstract

Coronary artery disease caused by atherosclerosis is a major cause of morbidity and mortality around the world. Data from preclinical and clinical studies support the belief that atherosclerosis is an inflammatory disease that is mediated by innate and adaptive immune signaling mechanisms. This review sought to highlight the role of Rac-mediated inflammatory signaling in the mechanisms driving atherosclerotic calcification. In addition, current clinical treatment strategies that are related to targeting hypercholesterolemia as a critical risk factor for atherosclerotic vascular disease are addressed in relation to the effects on Rac immune signaling and the implications for the future of targeting immune responses in the treatment of calcific atherosclerosis.

## 1. Introduction

The Rac family of small guanine triphosphate hydrolases (GTPases) is made up of a group of enzymes that hydrolyze guanosine triphosphate (GTP) in order to modulate many cellular functions, including motility, cell signaling, and cell cycling. Recently, the mechanistic impact of the Rac family in atherosclerosis has become increasingly appreciated in the context of understanding and developing potential treatments for coronary artery disease. In addition to placing Rac activity in the context of cell motility and proliferation, more recent studies have identified an atherosclerotic calcification signaling axis that is mediated by macrophage Racs and the potent inflammatory cytokine interleukin-1 beta (IL-1β). Moreover, there is a growing association between Rac expression/activity and progressive atherosclerosis in several experimental animal models. Additional studies indicated that current clinical treatment utilizing statins to target hypercholesterolemia has impacted Rac activity in a manner that alters the composition of atherosclerotic plaque beyond simply modulating cholesterol synthesis. The goal of this review was to outline our current understanding of the role of Racs in atherosclerotic vascular disease and to highlight areas where future research is needed. Determining the mechanistic basis of Rac immune signaling has the potential to provide valuable insight into the clinical treatment of calcific atherosclerosis.

## 2. Rac GTPases

The small guanine triphosphate hydrolase (GTPase) family of enzymes bind to and hydrolyze guanosine triphosphate (GTP), and are biologic modulators of a diverse array of cellular functions, including actin-based cytoskeletal networks and mobility, cell cycling, and signal transduction events that regulate RNA and protein biosynthesis [1,2,3,4,5,6,7,8].

GTPases are regulated in several ways, such as through the activation by guanine nucleotide exchange factors (GEFs), which help to release guanosine diphosphate (GDP) in order to bind and become active with GTP; inactivation by GTPase-activating proteins (GAPs), which expedite and exhaust the GTPase activity; and inhibition by GDP-dissociation inhibitors (Rho GDI), which keep the proteins in the cytosol and disallows activation by GEFs [9]. A well-known group of GTPases is the Rho family, which contains about 20 members and is broken down into six subfamilies [2,7,8]. The Rho family of small GTPases is a subgroup of the Ras superfamily and is known for its role in organelle development, cytoskeletal dynamics, cell motility, and other common cellular functions. Within the Rho family, there is a subset of proteins, namely, the Rac family, comprising four members (Rac1, Rac2, Rac3, and RhoG) [6]. Racs were initially discovered as substrates for ADP-ribosylation by Ras-related C3 botulinum toxin ADP-ribosyl transferase, with Rac1 and Rac2 being the most commonly studied members in the setting of inflammation [10,11]. Botulinum toxin is a classic endobacterial toxin that blocks the release of acetylcholine at the presynaptic terminal via NAD-dependent ADP ribosylation [12]. Subsequently, Racs were identified in several critical cellular functions, including mediating cell migration and motility, cytoskeletal projection and phagocytosis, cell division, axonal guidance, DNA synthesis, gene expression, superoxide production, and cellular transformation [2,3,6,7,8,9,13]. Several novel Rac functions were uncovered in the context of inflammation and vascular disease.

Racs are regulated by several different GEFs, including T-cell lymphoma invasion and metastasis 1 (Tiam1), triple functional domain protein (Trio), proto-oncogene vav (Vav1), guanine nucleotide exchange factor vav2 (Vav2), Rho guanine nucleotide exchange factor 2 (ArhGEF2), phosphatidylinositol 3,4,5,-trisphosphate-dependent Rac exchanger 1 protein (PRex1), and protein ECT2 (ECT2) [13,14,15,16,17,18,19]. Rac GEFs promote GDP dissociation from Rac to help facilitate GTP binding under a variety of different circumstances and within various subcellular locations, thereby promoting the diversity of functional downstream effects of GTPases within the cell. Racs are also inactivated by several different GTPase accelerating proteins (GAPs), including the Rho GTPase activating proteins (ARHGAP) ARHGAP10, ARHGAP15, and β2-chimaerin [20,21,22]. Rho-GDI inhibits Racs by sequestering them in the cytosol in a GDP-bound, inactive form so that they cannot exchange GDP for GTP. Rho-GDI may also be key in transporting Racs from the cytosol to the plasma membrane, where they can be activated by Rac GEFs (Figure 1) [23,24].

Four different Rac proteins have been described to date: Rac1, Rac2, Rac3, and RhoG. Rac1 is ubiquitously expressed by many different cell types and can be found in several tissues throughout the body, while Rac2 is primarily expressed in hematopoietic-derived cell types. Rac3 appears uniquely expressed in neurons of the ganglia and the central nervous system, and RhoG is expressed in fibroblasts, leukocytes, neuronal cells, and endothelial cells [6,10,19,25,26,27,28,29,30,31,32]. Racs are known as “small GTPases” due to their relatively small, 21 kDa size, and to maintain a functional GTP-binding domain, there is a limited number of sequence variations between the members. The Racs are known to bind to GTP via a standard GTP binding domain, which requires three highly conserved consensus sequences, namely, GDGAVGK (10-16), DTAG (57-60), TK(L/K)D (114-117). In the DTAG loop, G61 is a key residue in the GTP hydrolysis. The first two elements determine the interaction with the phosphate portion of the GTP molecule and the last element is involved in nucleotide specificity [7,33,34,35]. The Rac family, similar to Ras, also has a second guanine recognition site, namely, SAL, at residues 158-160 [35]. The sequence homology between Rac1, Rac2, and Rac3 is greater than 92% across the proteins, with the greatest divergence occurring in a short stretch of C-terminal residues (180-190) at a site that is termed the hypervariable region (Figure 2) [10,11]. Unlike the other Rac members, RhoG contains more sequence variation, with 72% sequence homology [31]. In general, Racs were often considered to have overlapping or redundant cellular functions, in large part due to the highly conserved sequence regions [19,28]. However, more recently, there has been increasing recognition that Racs may have varying functions, may cross-talk, or may even compete with each other for primary regulators, and that the dynamic changes in gene expression of one Rac can simultaneously influence the activation and function of another [36].

Post-translational modification, such as the addition of a lipid prenyl group or acyl chain, can affect the subcellular localization of GTPases [37]. Isoprenylation involves covalent modification with a C15 farnesyl or a C20 geranylgeranyl in which a thioester bond is formed at the carboxyl-terminal cysteine [38]. Farnesyltransferase recognizes the carboxyl-terminal CAAX box (A denoting aliphatic amino acids) (Figure 2), whereas geranylgeranyl transferase (GGTase 1) recognizes CAAL, which is a leucine terminal residue [39,40]. The terminal cysteine is essential for Rac geranylgeranylation, which helps with the localization to the inner plasma membrane or the nuclear envelope [26,41]. Rac1 and Rac2 are known to be able to translocate to the nuclear envelope, whereas Rac3 does not [41]. Rac3 localizes to the endomembrane at the cell periphery in lamellipodia and in the perinuclear region [25,42]. RhoG is activated in the perinuclear region [43]. Only a very small percentage of RhoG moves from the cytosol to the nuclear envelope with FITC-CTxB [44]. Consistently, RhoG was reported to move sequentially from the nuclear envelope to intracellular vesicles and the Golgi apparatus along the caveola/lipid raft-dependent endocytic route [43,44]. The involvement of RhoG in the caveola/lipid raft route seems highly specific and is not observed in the localization of other Rho/Rac family proteins. Other modification, namely, acylation and palmitoylation, involve a 16-carbon saturated fatty acid addition on a cysteine residue [40]. While palmitoylation is not essential for subcellular localization, the modification has some effects on membrane avidity and transforming activity through the support of nuclear localization [11,45,46]. This modification inhibits the binding of Rho-GDI and it may transpire at the terminal CAAX site, which is the same region that is involved in isoprenylation [41,45]. Notably, a prerequisite for the palmitoylation of Racs is isoprenylation [47].

The exact mechanisms of Rac nuclear localization are not well defined, but Rac1 was found to possess a nuclear localization sequence in its hypervariable region and the proteins are small enough (21 kDa) to passively diffuse through nuclear pores [48,49]. As mentioned, the cysteine site on Rac proteins is modified with a geranyl-geranyl group to promote translocation into the nuclear envelope [37]. Disruption of the prenylation prevents further modification and targeting, leading to cytosolic, inactive GTPases [5,11,37,41,50,51]. When treated with a GGTase inhibitor, there is decreased accumulation of Rac1 protein in the nucleus [52]. Membrane localization by cell fractionation was the primary technique used, which does not confirm active GTP-bound Rac. Current methods for identifying active GTP-bound Rac1 include PAK-binding domains and antibodies that are specific to the GTP-bound Rac [53,54]. In another study, it was found that mice with a GGTase-I deficit had high levels of GTP-bound Rac1 in the nuclear envelope of the GGTase-deficient macrophages [5,55], where GGTase I is a geranylgeranyltransferase, which is an enzyme that is involved in the protein prenylation of Rac1. The prenylation promotes Rac1 membrane association and stability. Therefore, GGTase I inhibitors impede Rac1 association with the nuclear envelope by disrupting its geranylgeranylation by GGTase I. Protein palmitoylation via palmitoyl acyltransferases (PATs) also influences Rac1 localization. The palmitoylation process involves the covalent attachment of fatty acid palmitate to cysteine, increasing Rac’s membrane association. Inhibitors of palmitoylation reduce Rac1’s normal localization to the cytoplasm, membrane, and nucleus, and increase localization to perinuclear space [56]. This interrupts typical Rac signaling and was proposed as a therapeutic target. Without palmitoylation, Rac is seen to localize to the endoplasmic reticulum and Golgi apparatus but not the plasma membrane [41]. The polybasic region upstream of the carboxyl terminus is also an important factor in iso-specific degradation [57]. The hypervariable region lies immediately upstream of the C-terminal cysteine and is often composed of a polybasic amino acid sequence. [11,26]. The electrostatic interaction between the polybasic region and the acidic phospholipids in the nuclear envelope appears to promote nuclear localization [58]. Racs have a proline-rich domain before the cysteine residue of the CAAX region that contributes to the translocation of the GTPase to form adhesion complexes with Rac-specific guanine nucleotide exchange factors (GEFs), which are located in subcellular regions, such as the plasma membrane, cytosolic vesicles, lipid rafts, and the perinuclear region [6,14,15,16,17,18,25,42,43,44,59]. The current model suggests that Rac1 is modified with a geranyl-geranyl group in the cytosol at the CAAX box in the polybasic region, resulting in the mobilization of Rac1 to form a complex with Rho-GDI with consequent shuttling, aided by karyopherin, from the cytosol to the nuclear envelope. It is currently unknown how Rac1 disassociates from Rho-GDI to interact with GEFs during the activation process [24].

The Rac proteins are well known for their effects on motility and cytoskeletal actions, such as lamellipodia formation, phagocytosis, actin motility, and adhesion [6,8,60]. They are also known to be key signal transducers of inflammation by promoting the production of reactive oxygen species (ROS) and by effecting changes in gene expression, particularly of growth factors and cytokines, via the activation of downstream transcription factors [1,2,3,4,5,6,8,61,62,63,64]. Under normal conditions, Rho-GDI is bound to GDP-Rac1 in the cytosol, holding it in its inactive state [1]. When active, Rac1 mediates gene transcription through transcription factor NF-kB, Jun N-terminal kinases (JNKs), and p38 mitogen-activated protein kinases (MAPKs), which promote activator protein-1 (AP1) transcription factors [65]. Through these transcription factors, Rac1 may influence downstream proteins that affect the cell cycle and G1/S progression [61,62,63,64]. Notably, through the activation of NF-kB, Rac1 also appears to activate the transcription of proinflammatory cytokines, such as TNF-α and interleukin-1β (IL-1β) [28,66]. Rac1 is critical for cell motility and migration and is responsible for lamellipodia protractions via its regulation of p21-activated kinase-mediated polymerization [60,67]. Rac1 is traditionally believed to be active in the plasma membrane to facilitate this function, and high concentrations of active Rac1 were observed where the lamellipodia are generated [68]. Rac1 is not required for the basic functions of all cell types, as some cell types have evolved redundancy or alternative mechanistic pathways for similar functions [69,70]. For example, neutrophils, but not macrophages, require Rac1 for migration. Neutrophils are dependent on Rac1 and Rac2 to facilitate migration, and they are 20 times faster than macrophages [70,71,72]. Rac1 also participates in the formation of ROS via the activation of NADPH oxidase. NADPH oxidase interacts with the extracellular oxygen species to generate a superoxide anion in the intracellular environment [62,73]. The assembled and active NADPH oxidase complex consists of a plasma-membrane-bound flavocytochrome b558 (gp91phox/p22phox), three phosphorylated cytosolic subunits (p47phox, p40phox, and p67phox), and an active Rac-GTPase [62,64,74,75]. The C-terminal polybasic motif, namely, KKRKRK, is directly responsible for Rac1’s interaction with NADPH oxidase complex, which is found inside the nuclear envelope [35]. This motif is within the hypervariable region, which can account for the differences in the respective roles of various Rac family members in ROS production. It is also known that post-translational isoprenylation is important for Rac1’s interaction with NAPDH oxidase because when isoprenylation is inhibited, NADPH is also inhibited [76]. Rac1 is primarily associated with macrophage NADPH oxidase formation, while Rac2 is primarily associated with neutrophil NAPDH oxidase [69,72]. There are also important relationships between the Racs and intracellular membrane trafficking; ROS production; nuclear transcription factors, such as NF-κB and the cell-cell and cell-extracellular adhesion matrix; and cell growth and survival, but there is more research to be done [2,3,5,6,8,9].

Rac2 is important for chemotaxis, superoxide production, and phagocytosis in hematopoietic cells. Rac2 is a major regulator of IL-1β expression [28], in part via the suppression of Rac1’s activation of NF-kB and production of ROS [62,65,73,75]. The distinct hypervariable region before the carboxyl-terminal cysteine determines the specificity of Rac2′s function in neutrophil chemotaxis and superoxide generation, which differs from Rac1’s function [26]. The hypervariable region also allows Rac2 to translocate to certain areas at the nuclear envelope [41]. The specificity of Rac2 in superoxide production in neutrophils is determined mainly by the RQQKRA polybasic residue. The specificity of neutrophil chemotaxis in Rac2 is due to the polybasic region and aspartic acid 150. D150 specifies the appropriate polarization of cortical F-actin in the correct direction of migration [5,26]. Rac2 is required for neutrophil migration and NADPH oxidase formation [69,70,72,73]. Rac2 is an important modulator for neutrophils, as supported by the fact that patients with acquired mutations in Rac2 have increased postnatal infections and neutrophil dysfunction, hence requiring a hematopoietic stem cell transplant [77].

Rac3 is primarily expressed in the central nervous system and neurons [6,7,8]. Rac1 and Rac3 appear to have opposing effects on cell adhesion and morphology [42]. In neurons, Rac3 is responsible for cell rounding and cell adhesion, whereas Rac1 is responsible for cell migration and motility. This is also likely due to the differences that are conferred by their respective C-terminal hypervariable regions. For example, Rac1 enters the cell’s nuclear envelope with its hypervariable region nuclear localization sequence, whereas Rac3 is primarily found in the perinuclear region and appears unable to localize into the nuclear envelope [42]. When mutated to have the same hypervariable region as Rac1, Rac3 is able to localize to the nuclear envelope and effectively mimic Rac1’s functions in neurons [42].

RhoG is the most recently discovered member of the Rac family and has not yet been studied as extensively as the other members. As mentioned, RhoG shares about 72% sequence homology with Rac1, and like the other Racs, RhoG has an identical GTP binding domain [6,15,31]. RhoG is primarily expressed in fibroblasts, leukocytes, neuronal cells, and endothelial cells, and has similar functions of cell motility and cytoskeletal changes [5]. The unique N-terminal extension of Rho-GDI, which encompasses a predicted α-helix, is necessary for its localization at the Golgi and is expected to be pivotal for the down-regulation and/or trafficking of RhoG inside the cell [43]. Shuttling of the Rho-GDI/RhoG complex may follow a vesicular route rather than, or in parallel to, cytoplasmic diffusion of a soluble Rho-GDI/Rac complex. The movements along microtubules would be consistent with previous observations that microtubules are necessary for RhoG functions [43]. RhoG triggers fibroblasts to form lamellipodia and filopodia [31]. It was demonstrated that constitutively active mutations of RhoG can induce the formation of membrane ruffles and filopodia in the fibroblasts [31]. RhoG preferentially localizes to cytoplasmic vesicles, far away from the plasma membrane, but when mutated to have the same hypervariable region as Rac1, RhoG is able to translocate the plasma membrane from the cytoplasm [5].

There are currently several gene-deletion mouse models of the Rac family, specifically *Rac1*, *Rac2*, and *Rac3*. Rac1, Rac3, and RhoG have 100% sequence homology between the human and mouse proteins, whereas Rac2 has 99% sequence homology [5]. Overall, the sequence homology supports the potential translatability of Rac mouse model findings to a human model. Of note, heterozygous *Rac* deletion models still have 50% function, supporting the belief that Racs are allele-dose dependent [78]. Global *Rac1*-deletion mice are embryonically lethal and demonstrate a range of defects in germ-layer formation; therefore, tissue-specific knockout models have been widely used to study Rac1 function [6]. *Rac2-* and *Rac3-*deletion mice do not show obvious developmental defects, but have cell-type-specific functional defects and demonstrate pathology upon challenge [5,6]. Of note, human neutrophil immunodeficiency syndrome is very similar to a *Rac2*-deletion model in mice and their neutrophil profiles are very similar [77]. Human neutrophil immunodeficiency syndrome is a primary immunodeficiency that is characterized by neutrophilia with severe neutrophil dysfunction, namely, leukocytosis, which gives a predisposition to bacterial infections, and poor wound healing. In macrophages, Rac1 is the more prevalent isoform and *Rac2* deletion has only a minor effect on migration, suggesting Rac1 may compensate to some degree for the loss of Rac2 via a redundant role [19,28]. In the setting of inflammatory stimuli, macrophages increase Rac1 expression by 2-3-fold [19]. In the setting of *Rac2* deletion, the levels of GTP-bound Rac1 are increased for every given inflammatory stimulus [28]. The loss of Rac2 appears to increase the availability of Rac1 for its respective GEFs, leading to the consequently increased activity [28].

Constitutively active forms of Rac1 include the splice variant Rac1b (Figure 3), mutant Rac1(Q61L), and mutant Rac1(G12V) [79,80,81]. In Rac1(Q61L) and Rac1(G12V), the amino acid substitutions prevent the GAP-stimulated GTPase activity of Rac1, thereby maintaining an active form, namely, Rac1-GTP. They are unaffected by Rho GDP dissociation inhibitor (Rho-GDI), which is an inhibitory regulator of Rac1 [23]. Both the Q61L and the G12V mutants are therefore commonly used when studying a GTP-bound, constitutively active state of Rac1. The Rac1b splice variant is an alternatively spliced isoform of Rac1, with a 19 amino acid insertion (exon 3b), causing increased GDP/GTP exchange and impaired GTP-hydrolysis [82]. It was indicated as an important stimulant in transcription and decreased adhesion of breast and colon cancer cells. Its overexpression results in increased TCF (T-cell factor)-mediated gene transcription, whereas suppression results in decreased expression of the Wnt target gene cyclin D, which is important in the regulation of cell cycle transitions and migration of macrophages [83,84]. Rac1b as a constitutively active form of Rac1 is therefore relevant in pathogenesis involving macrophage Rac signaling, such as in the case of atherosclerosis. Further investigation of constitutively active Rac mutants may give additional insight into the role of Racs in atherogenesis.

## 3. Overview of Atherosclerosis

Atherosclerotic heart disease is the leading cause of morbidity and mortality in the world [85]. In the United States, for men and women between 60 and 79 years of age, coronary artery disease occurs at 24 and 15% prevalences, respectively, and for those over 80, this increases to 36 and 24%, respectively [86]. Hyperlipidemia is a key risk factor for atherosclerosis since disease progression involves the development of lipid-rich plaques in the intimal layer of arterial blood vessels [87]. These plaques are defined as accumulations of abnormal material in the arterial vessel wall, such as lipids, cellular debris, fibrotic tissue, and deposits of calcium [87]. In the formation of a plaque, extracellular lipids assemble in the intimal layer under the endothelial monolayer [88]. ROS plays a role in lipid peroxidation of native low-density lipoprotein (LDL) that has localized from the vessel lumen to the subendothelial space [89]. Once recruited to the site of inflammation, macrophage scavenger receptors, such as the class A (SR-A) or class B (SR-B) receptors, help macrophages to recognize and phagocytose the oxidized LDL (oxLDL) [90,91]. Macrophages may become oversaturated by the amount of phagocytosed oxLDL, resulting in transformation into “foam cells,” which are loaded with lipids and proliferate proinflammatory signaling and recruitment of additional myeloid cells. The “foam cell” name is derived from its foamy appearance under the microscope due to the LDL-rich nature of the macrophages in cytoplasmic droplets [92]. Foam cells proliferate the development of cholesterol microcrystals. These microcrystals cause lysosomal instability and activate the NOD-, LRR-, and pyrin-domain-containing protein 3 (NLRP3) inflammasome, which is known to increase the secretion of the mature proinflammatory cytokine, namely, interleukin-1β (IL-1β), into the extracellular plaque region [93,94,95]. IL-1β is a key cytokine that recruits additional pro-inflammatory cells and stimulates the downstream production of IL-6 and C-reactive protein (CRP) in the liver [96]. With increased inflammatory signaling, the growing plaque further accumulates cholesterols, lipids, endothelial cells, mesenchymal cells, immune cells, and cellular debris, which proliferate the inflammatory state of the arterial vessel wall and contribute to atherogenesis.

Atherosclerosis involves an accumulation of plaques, which narrow the arteries and can cause cardiovascular complications, such as coronary artery disease (CAD), peripheral artery disease (PAD), and cerebrovascular disease (CVA) or stroke [87]. Constriction of the arterial lumen can impede myocardial oxygen supply, alter blood pressure, induce cardiovascular stress, and can result in thrombotic obstruction or myocardial infarction (MI) due to plaque rupture. Cardiac computed tomography (CT) evaluates calcification within the plaques and helps to determine the atherosclerotic burden, which is clinically used as a predictive value of all-cause mortality [97,98]. The composition of calcium within plaque may confer varying degrees of stability or risk of plaque rupture, and thus calcification can be an important biomarker of the event risk [99]. One type of calcification, namely, microcalcification, involves <50 μm calcium nodules that are deposited by vascular smooth muscle cells (VSMCs) that differentiate into early-phase osteoblasts, as well as vesicle mineralization from apoptotic M1 macrophages and VSMCs [100,101]. The other type of calcification, namely, macrocalcification, involves ≥50 μm calcium nodules that are deposited by the M2 macrophage mediation of VSMCs and osteoblast differentiation [101]. Microcalcification is termed “spotty” calcification due to its spotty deposit nature and results in increased mechanical stress on the fibrous cap of the plaque, leading to a greater chance of rupture. On the other hand, since macrocalcification involves dense, stable, macroscopic deposits of calcium, there may be a decrease in the risk of rupture and adverse cardiovascular events [102]. While macrocalcification was indicated to have a stable phenotype against plaque rupture, additional studies may clarify the potential risk of an aortic dissection that is associated with macrocalcification [100,103,104]. The question of whether micro- or macrocalcification influences the stability and vulnerability of plaques is an important one that requires further investigation in the context of the molecular mechanisms leading to atherosclerotic calcification and clinical outcomes. Since the rupture of an atherosclerotic plaque contributes to myocardial infarction and stroke, which are two major causes of mortality around the world, it is important to isolate the molecular mechanisms that drive the disease state and plaque formation [105,106,107].

Inflammatory cytokine IL-1β is a key element in inflammatory signaling and atherosclerosis [108]. Clinical data from atherosclerotic patients showed increased IL-1β protein and mRNA levels in endothelial cells and macrophages, with increased levels corresponding to the increased severity of atherosclerosis [109,110]. ProIL-1β is a 33 kDa precursor that is cleaved by caspase-1 into the mature 17 kDa IL-1β form, which exerts signaling effects on endothelial cells, smooth muscle cells (SMCs), monocytes, and macrophages. IL-1β promotes an inflammatory response in endothelial cells that increases the levels of adhesion molecules (intercellular adhesion molecule-1 (ICAM-1), vascular cell adhesion molecule 1 (VCAM-1), monocyte chemoattractant-1 (MCP-1)), chemokines, and cytokines in the arterial intima, which further proliferates recruitment of proinflammatory cells to the site of inflammation and promotes the differentiation of VSMCs [111,112]. VSMCs that were treated with recombinant IL-1β demonstrated increased expression of G protein-coupled receptor P2Y_2_R, which induced intracellular calcium mobilization, PKC and PI3K activation, and VSMC proliferation [113]. In an apolipoprotein E-deficient (*ApoE*^−/−^) mouse model of hyperlipidemia, treatment with recombinant IL-1β increased the VSMC transcription of osteogenic factors, such as RUNX2, MSX2, SOX9, and OSX, and in a cell culture matrix, IL-1β increased calcium deposition into the extracellular matrix in a dose-dependent manner [28]. While previous studies attribute increases in osteogenic gene programming to TNF-α, recent work has uncovered that IL-1β is a key cytokine in the expression of these factors [114,115,116]. Recent data suggested that IL-1β that is produced in response to TNF-α by the VSMCs themselves contributes to fibrous cap formation in late-stage atherosclerosis in an *ApoE*^−/−^ model [117,118]. The proposed mechanism suggests that SMC-specific IL-1β promotes beneficial outward remodeling of the lesion. Further research should delineate the contributions of VSMC- and macrophage-specific IL-1β on atherogenesis and the progression of the disease. Interleukine-6 (IL-6) is another critical inflammatory cytokine that is upregulated by IL-1β-mediated transcription in macrophages. Higher circulating levels of IL-6 lead to the increased expression of acute-phase reactants by the liver, including CRP, as well as fibrinogen and plasminogen activator inhibitor-1 (PAI-1), which are clotting factors that may increase risk of thrombosis. IL-1β is therefore an upstream regulator of CRP in the liver through the regulation of IL-6 [119].

VSMC proliferation and migratory phenotypes were also examined in the context of Rac signaling. SMC migration in response to platelet-derived growth factor (PDGF) is a key event in the pathology of atherosclerosis and is dependent on Rac signaling [120]. The inhibition of Rac shows significantly reduced PDGF-induced and ROS-sensitive VSMC migration. Rac GEFs, such as Kalirin, also play a role in promoting the Rac1-GTP modulation of SMC migration and proliferation [121]. The *ApoE*^−/−^ mouse model demonstrated an upregulation of arterial Kalirin in early atherogenesis, and Kalirin loss-of-function via gene deletion, RNAi, and inhibitor treatment showed reduced Rac1 activation, as determined by Rac-GTP levels and reduced SMC proliferation and migration. VSMC differentiation toward a contractile phenotype is dependent on the relative Rac and Rho levels in response to sites of injury and modulation of arterial extracellular matrix (ECM) stiffness [122]. A large relative increase of Rac in response to initial injury supports VSMC proliferation and de-differentiation, which is a phenotype that is associated with arterial stiffening in atherosclerosis, whereas a subsequent relative increase in Rho may induce VSMCs to a differentiated phenotype during injury resolution. Changes in VSMC composition and phenotype are dynamically dependent on changes in the microenvironment, and Rac signaling from surrounding macrophages and endothelial cells may uniquely impact vascular pathophysiology.

## 4. Rac Observations in Inflammatory Atherosclerosis

We previously identified an opposing relationship between Rac2 and Rac1 in pro-inflammatory macrophages [28]. Upon *Rac2* deletion, primary macrophages demonstrated consequent increases in Rac1 activity along with elevations in mature IL-1β secretion in response to inflammasome stimuli. The pattern of Rac1 expression was not different between bone-marrow-derived macrophages (BMDMs) from wild-type and *Rac2*-deleted mice; however, active Rac1 (GTP-bound) was significantly increased in BMDMs from *Rac2*-deleted mice, as measured by both p21-binding domain (PBD) pulldown affinity assay. This suggested a competition between Rac1 and Rac2 for a similar GEF, whereby the *Rac2* gene deletion resulted in increased macrophage Rac1 activity. The elevations in IL-1β were suppressed by inhibitors for NF-kB and ROS production, indicating a potential role for both as downstream effectors of Rac1 in the expression and activation of IL-1β.

Given the background of the *ApoE*-deficient mouse model of hyperlipidemia, *Rac2* deletion resulted in elevated circulating IL-1β in blood plasma from mice on a cholesterol-supplemented high-fat diet. The increased macrophage Rac1 activity and consequent elevations in IL-1β led to increases in atherosclerotic calcification in the aortas of the *Rac2*-deleted mice. Treatment with an IL-1 receptor antagonist (IL-1ra, anakinra) mitigated the progression of calcification, supporting its IL-1β dependence. In sum, the data support an antagonistic relationship between Rac2 and Rac1 in mediating IL-1β expression and consequent IL-1β-dependent vascular remodeling. However, more studies are required to define the exact mechanisms of Rac1-dependent IL-1β expression, along with the relative contributions of NF-kB and ROS production as potential downstream effectors of Rac1. Moreover, outside of *Rac2* deletion leading to progressive atherosclerotic calcification, the relative impact of this Rac1–IL-1β signaling axis in the natural progression of atherosclerosis and/or atherosclerotic calcification in the context of hyperlipidemia remains unclear.

Prior studies that assessed the role of Rac1 in atherosclerosis progression were carried out using the Watanabe heritable hyperlipidemic (WHHLMI) rabbit model [123]. The studies evaluated Rho and Rac1 GTPase activation regarding lipid modification geranylgeranylation, GDP/GTP exchange or GTP-loading, and protein membrane translocation. These indicators of Rac1 activation were used to determine the relative protein mediation in atherosclerotic plaques of the WHHLMI animals. The investigators also noted an elevated expression of Rac1 in association with smooth muscle cells and macrophages in the histological analysis of the animal aortic atherosclerotic lesions, which was higher in advanced 7-month aortas compared to 3-month aortas. They observed increased activity of Rac1 in the 7-month atherosclerotic aortas using a PBD pull-down assay to show increased GTP-loading of Rac1, corresponding with advanced plaques. This observation was accompanied by significantly greater membrane-bound RhoA and Rac1 in the 7-month atherosclerotic aortas compared to the 3-month aortas, and yet somewhat paradoxically, the investigators identified reduced geranylgeranylation of Rac1. In vitro experiments with cultured rabbit smooth muscle cells supported oxidized LDL (oxLDL) influence on increasing Rac1 protein levels. The increased Rac1 expression from atherosclerotic aorta tissue and oxLDL-treated smooth muscle cells differed from our findings in the *ApoE*^−/−^ mouse model, where Rac1 expression remained constant relative to Rac2 in the atherosclerotic aortas over time and in inflammasome-stimulated smooth muscle cells [28]. However, we did find that inflammasome stimulation of macrophages led to modest increases in Rac1 expression overall.

More recently, an immunofluorescent investigation of human carotid atherosclerotic plaques demonstrated increased macrophage Rac1 expression in advanced plaques than intermediate plaques, though it is unclear whether this represents increased inflammatory cell intimal infiltrate or increased gene expression [124]. This observation is comparable to the WHHLMI rabbit model and our findings of inflammasome stimulation of macrophages but did not assess the activation state of Rac1 or its association with plaque features (i.e., immune cell infiltration, necrosis, thrombosis, or calcification) [28,123]. These differences in Rac1 expression levels between the studies may reflect differences that are intrinsic to the animal models or intrinsic to the treatments lipopolysaccharide plus cholesterol crystals vs. oxLDL. These studies support a relationship between models of progressive atherosclerosis and increased Rac1 activation, yet they fall short of drawing a mechanistic relationship between Rac1 and progressive calcification.

Another prior study evaluated Rac1 in the context of chronic infection with *Chlamydophila pneumoniae*, which is a Gram-negative bacterium that is associated with asthma and atherosclerosis development [125]. The investigators demonstrated greater levels of mature IL-1β in *Chlamydophila*-*pneumoniae*-infected human monocytes. As previously mentioned, NLRP3 is responsible for cleaving IL-1β into its active form by converting pro-caspase-1 into its active form, namely, caspase-1, which in turn cleaves proIL-1β into mature IL-1β. The knockdown of NLRP3 and caspase-1 in human peripheral blood mononuclear cells reduced IL-1β levels in cells infected with *Chlamydophila pneumoniae*, confirming the inflammasome as a mechanistic precursor for mature IL-1β development in this chronic infection. Additionally, the authors determined that that Rac1 operates upstream of NLRP3, caspase-1, and IL-1β. After treating human monocyte cells with a Rac1 inhibitor, the investigators found that both IL-1β levels and caspase-1 activation decreased in a dose-dependent manner. Interestingly, experiments involving the depletion of Rac1 mRNA with Rac1 siRNA decreased mature IL-1β protein levels, but not necessarily the creation of proIL-1β mRNA. Though further study is needed, this suggested that Rac1 may play an important and underappreciated role in the post-translational regulation of IL-1β maturation/secretion. Once the mature form of IL-1β is secreted in the cell, it can perform extensive atherogenic signaling, recruit additional proinflammatory cytokines, and promote self-expression via IL-1 receptor NLRP3 priming within macrophages [108]. While the model of *C.*-*pneumoniae*-infected human monocytes gives insight into potential mechanistic patterns, additional experiments are required to assess the pattern of Rac1-dependent IL-1β expression in a preclinical model of aseptic hyperlipidemia.

An additional analysis utilized the PCSK9 protein, which suppresses the number of surface LDL receptors, leading to hyperlipidemia and consequent atherosclerosis [124,126]. Using a constitutive myeloid (LysM Cre recombinase) *Rac1*-deficient model that was infected with adenovirus overexpressing protein convertase subtilisin/kexin type 9 (AdPCSK9) combined with a 20-week high fat diet, the investigators demonstrated that *Rac1*-deficient macrophages had modest reductions in the inflammatory cytokines IL-6 and TNF-𝛼 in mouse serum and cell-cultured bone-marrow-derived macrophages [124]. Unfortunately, the IL-1β expression was not assessed. The *Rac1*-deficient macrophages also demonstrated less foam cell formation, as evaluated using Oil Red O staining, and the AdPCSK9-treated animals on a high-fat diet demonstrated mild reductions in aortic lipid burden, as measured using enface Oil Red O staining. The observational findings in human atherosclerotic plaques, in combination with the outcome of their in vivo transgenic mouse data, suggest that Rac1 expression plays a role in atherosclerotic progression; however, the underlying mechanism of action remains unclear. Results from this study should be replicated to confirm the findings and further elucidate a mechanism behind the proposed observations.

## 5. Current Therapy: Statins

3-Hydroxy-3-methylglutaryl-CoA (HMG-CoA) reductase inhibitors, or statins, are the most widely used medications in the primary and secondary prevention of patients with atherosclerotic vascular disease [127]. Statins act by competitively inhibiting the active site of the HMG-CoA reductase enzyme, impeding the conversion of HMG-CoA into mevalonate, an important step in the cholesterol synthesis pathway (Figure 4). Surface LDL-receptors, which are responsible for the hepatic incorporation of LDL-C from the blood, are upregulated in the case of this decreased cholesterol synthesis, leading to increased uptake of LDL-C from the blood. Abundant clinical evidence supports statin success in lowering LDL-C to reduce coronary mortality and the risk of cardiovascular events in long-term studies by upward of 30% [87]. Furthermore, statins were found to improve endothelial dysfunction, increase nitric oxide bioavailability, have antioxidant effects, inhibit inflammatory responses, and stabilize atherosclerotic plaques [128]. In fact, statins have been so successful in clinical practice that it is widely recognized that they reduce events out of proportion to the cholesterol-lowering effects. These “pleiotropic” effects of statin therapy may come from additional effects beyond simply inhibition of cholesterol synthesis, including the isoprenyl-based posttranslational modification of Racs [129,130,131,132]. Statin treatment influence on atherosclerotic calcification has been controversial, as early studies suggested that statins reduce atherosclerotic calcification and more recent studies found the opposite [133,134,135,136]. Moreover, the recent studies indicate a correlation between statin therapy and increased relative plaque calcium to lipid composition, but the mechanism driving this observation remains undefined [134,137,138]. Histological analysis of more than 830 statin patients demonstrated that a high dosage of statins may significantly reduce plaque volume and increase the dense calcium volume when compared with a lower dosage of statins [139,140]. Investigation of the plaque composition shows that high-dose statin therapy increases calcium volume while leaving fibro-fatty necrotic core volumes unchanged. The findings suggest that statins affect the stability of the plaques by modulating the lipid-to-calcification burden, perhaps by decreasing the plaque risk of rupture and erosion. Statins are known to inhibit the synthesis of isoprenoid intermediates, such as farnesyl pyrophosphate, geranylgeranyl pyrophosphate, isopentanyl adenosine, dolichols and polyisoprenoid side chains of ubiquinone, heme A, and nuclear lamins; the first two are important activators of Rho and Rac proteins, as indicated above, and recent studies have assessed the impact of statin therapy Rac-mediated atherosclerotic calcification [141].

## 6. Rac and Immune-Targeted Treatments

As mentioned above, several studies have noted an association between Rac expression and the progression of atherosclerosis. Targeting Rac1 as a therapeutic strategy is logical given its role in activating NADPH oxidases, such as NADPH oxidase 2 (NOX2), which is responsible for subsequent ROS production, thereby promoting atherogenic oxidative stress [142,143,144]. Furthermore, Rac signaling is also responsible for the movement of smooth muscle cells and leukocytes into the arterial inner layer, which are accumulations that correspond with atherosclerotic plaque development [13]. Based on these observations, there are two drug classes for mitigating Rac signaling through the targeting of Rac lipid modification, which includes statins and GGTase I inhibitors (GGTIs) [145].

Research is uncovering the role that statins have in modulating the GTPase and Rac1 signaling pathways in human smooth muscle cells [146]. A study of statins’ impacts on Rho protein localization found that treatment with simvastatin impeded the Rac1 carboxyl-terminal methylation that is required for proper protein localization. Accordingly, simvastatin was found to impede Rac1’s downstream matrix metalloproteinase-1 (MMP-1) generation and GTP-loading, as evaluated by immunoblotting of total cell lysate, consequently leading to decreased collagen breakdown and mitigation of Rac1-mediated smooth muscle cell migration in culture.

Another study observed a statin effect on Rac1 that was distinct from the prenylation mechanism. The investigators noted that treatment using atorvastatin and pitavastatin induced intranuclear Rac1 degradation in cultured human umbilical vein endothelial cells (HUVECs), as evaluated using nuclear and cytosolic Rac1 expression via Western blot analysis [147]. Though nuclear Rac1 signaling was indicated in the disease progression of certain cancers and tumors, more research needs to evaluate the role of nuclear Rac1 expression in atherosclerosis [148]. An important consideration should be noted that statins suppress signaling pathways of many GTPases aside from Rac1, including RhoA, which may have alternate effects [149]. Therefore, the statins’ impacts on GTPase prenylation is a summation effect, and it is not yet clear how the proposed statin-induced nuclear degradation of Rac1 contributes to the overall statin phenotype.

Our group investigated an alternative statin impact on Rac1 signaling. Using a combination of in vitro methods and experimental mouse models, we determined that atorvastatin treatment increased Rac1 activation in primary monocytes and macrophages. Atorvastatin disrupted the inhibitory complex between Rho GDP dissociation inhibitor (Rho-GDI) and Rac1, thereby releasing more Rac1 into the cytosol from its inhibitory pool and, consequently, increasing the levels of GTP-bound Rac1. This finding is seemingly contradictory to some previous results, indicating that statins may be atheroprotective by inhibiting Rac1 [130,150,151]. In our hands, the macrophage-specific active GTP-Rac1 increased mature IL-1β, leading to increased calcification of atherosclerotic plaques (Figure 5) [28]. This is interesting in the context of recent evidence showing that statins promote atherosclerotic plaque stability but also may actually increase the density of calcium deposition [102,140]. This study on macrophage-specific Rac1 presents an actual mechanism whereby statins influence the composition of calcification within atherosclerotic plaque, raising many questions about what it means to have atherosclerotic calcification in the context of statin use. Further investigation should evaluate the summation effect of statins on alternate GTPases, as well as determine the mechanism by which statins increase IL-1β expression.

There may also be a role for GGTase I inhibitors and palmitoylation inhibitors in targeting Racs in the context of atherogenesis. Upon conditional deletion of GGTase I in mouse macrophages, there is an increase in proinflammatory response demonstrated by increased cytokine production in atherosclerosis and rheumatoid arthritis mouse models [152,153]. Subsequent deletion of Rac1 reverses the inflammation associated with GGTase I-deficient mice. However, despite this increase in inflammatory signaling in the macrophages, GGTase-I-deficient mice (*Pggt1b*Δ/Δ) demonstrated a significant reduction in the atherosclerotic lesion area. Therefore, GGTase I inhibitors may play a role in reducing Rac- and Rho-driven atherosclerosis [153]. However, neither palmitoylation inhibitors nor GTTase I inhibitors have been extensively researched in a clinical setting, and mechanisms of action for both families of inhibitors are not necessarily specific to Rac, given their impact on several types of small GTPases.

## 7. Conclusions

Atherosclerosis is a chronic inflammatory immune-mediated disease, and treatment of atherogenesis should reflect our current knowledge of immune-related pathogenesis. The off-target mechanisms of statins, which are the most widely used medication in atherosclerotic treatment, are still being researched. Though statins, as HMG-CoA reductase inhibitors, lower LDL-C, there is growing evidence to support alternative effects on immune mechanisms. Several studies documented the association of Rac levels and activity with the progression of atherosclerosis, but further research needs to elucidate the exact mechanisms and biological components of the atherosclerotic process that Racs influence. Some studies showed that Rac1 inhibition decreased atherosclerosis in animal models via improved endothelial function. However, other data suggest that Rac signaling influences IL-1β signaling and the activation of inflammatory macrophages to progress atherosclerotic calcification. There is now growing data showing that statins activate rather than inhibit Rac1 in these inflammatory settings. Ultimately, therapeutic targeting of Racs will require careful consideration of their wide array of effects and diverse influence in signal transduction events. Moreover, targeting Racs in atherosclerosis must take into account their cell- and tissue-specific effects and the varying impact of each Rac across the multiple biological processes that drive complex vascular disease.

## Figures and Tables

**Figure 1 cells-10-02808-f001:**
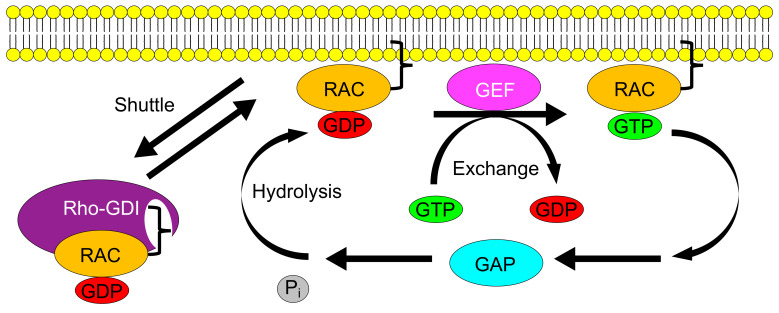
Illustration of the Rac activation cycle. Racs are active in their GTP-bound state and inactive when bound to GDP. The vast majority (90–95%) of Rac proteins are inactive and sequestered in the cytosol by Rho-GDI. Activation begins with the shuttling of Racs to the plasma membrane, where they can be activated by the GEF-mediated exchange of GTP for GDP. GAPs facilitate the hydrolysis of GTP to GDP and inorganic phosphate, leading to inactivation. Isoprenylation (black bracket) anchors Racs to the plasma membrane but also plays a role in Rho-GDI binding so that Rho-GDI can also shuttle inactive Racs away from the plasma membrane.

**Figure 2 cells-10-02808-f002:**
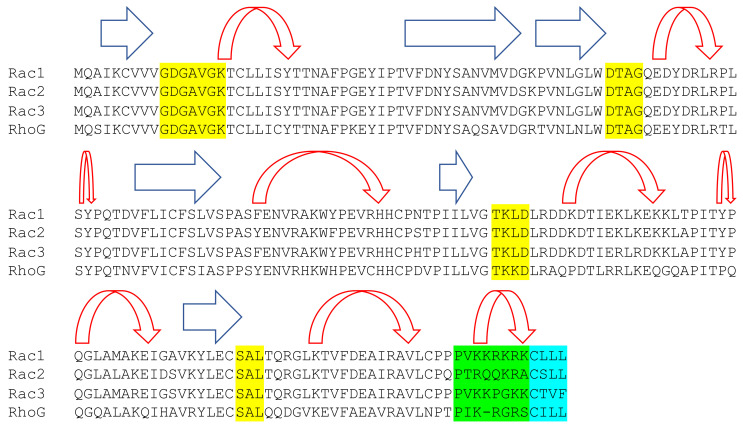
Rac family proteins demonstrate 92% sequence homology between Rac1, Rac2, and Rac3 and 72% homology between RhoG and the other Racs. Human sequence alignments for Rac1, Rac2, Rac3, and RhoG. Secondary structures are illustrated above the sequences with beta sheets denoted by blue arrows and alpha helixes denoted by red arrows. GTP binding and hydrolysis domains (highlighted in yellow) are 100% conserved. The hypervariable region is highlighted in green. The CAAX box (light blue) is recognized by geranylgeranyl transferase.

**Figure 3 cells-10-02808-f003:**
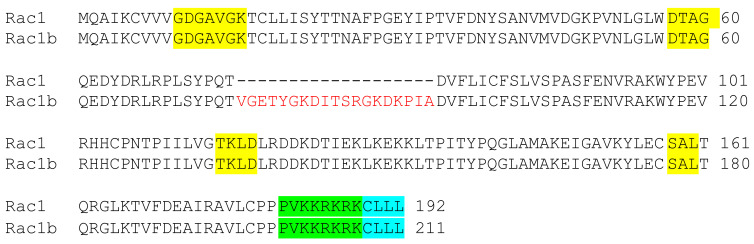
Splice variants Rac1 and constitutively active Rac1b differ by a 19-amino-acid insertion in Rac1b. Sequence alignments of Rac1 and alternatively spliced isoform Rac1b. GTP binding and hydrolysis domains (highlighted in yellow), GTP binding domain (yellow), hypervariable region (green), and AAX box (blue). The 19-amino-acid insertion highlighted in red leads to increased GDP/GTP exchange and impaired GTP-hydrolysis.

**Figure 4 cells-10-02808-f004:**
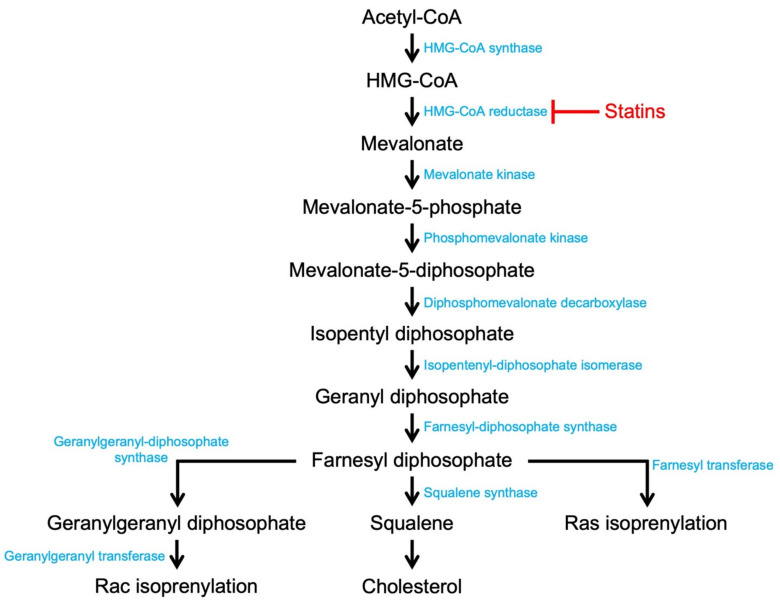
Statins disrupt the mevalonate pathway and consequent Rac isoprenylation. Illustration of the mevalonate pathway leading to the production of cholesterol and isoprenoids. Farnesyl diphosphate (FPP) and geranyl-geranyl diphosphate (GGPP) post-translationally modify the signaling proteins Ras and Rac, respectively. Statin inhibition (red) of HMG-CoA reductase disrupts both cholesterol synthesis and Rac isoprenylation upstream of the independent branch point of farnesyl diphosphate.

**Figure 5 cells-10-02808-f005:**
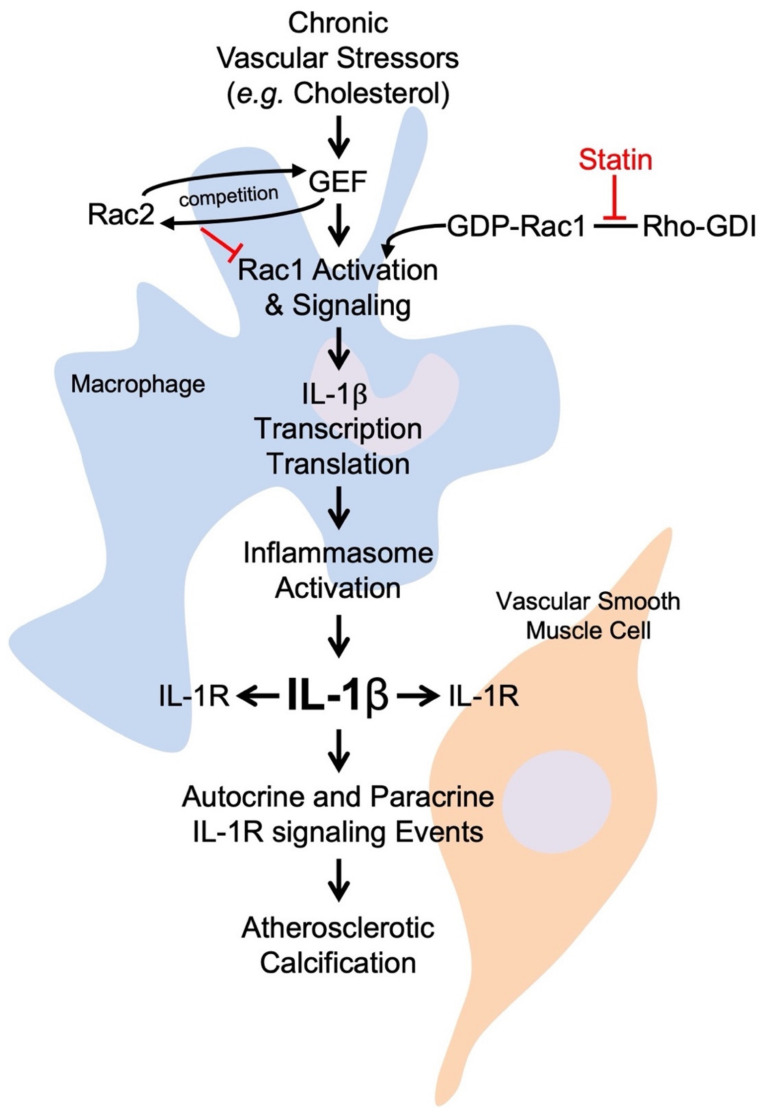
Working model of how Racs mediate IL-1β expression and the consequent atherosclerotic calcification. Vascular stressors activate innate inflammatory pathways in macrophages, leading to Rac1 activity. Rac2 competes with Rac1 for common GEFs, tempering Rac1 activity for any given stimulus. Rac1 activity is associated with elevated IL-1β expression, though the underlying mechanisms are not yet fully defined. Increased IL-1β expression and the consequent IL-1β–IL-1R signaling promotes atherosclerotic calcification by activating osteogenic gene programs in vascular smooth muscle cells. Statin treatment disrupts the complex between Rac1 and Rho-GDI while also allowing for increased Rac1 activation and Rac1-mediated IL-1β expression.

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
