# Peer review of "Rac GTPase Signaling in Immune-Mediated Mechanisms of Atherosclerosis"

_cells, 2021, doi:10.3390/cells10112808_

Round 1

Reviewer 1 Report

With this work Lee and colleagues gives a detailed descriptions of the RacGTPases and their involvement in the immune-mediated atherosclerosis.

Introduction should be differently structured because as is it risk to lose the reader, too long.

A brief introduction should first introduce Rac family and place them in the context of atherosclerosis giving an overview of what will be the scope of the review.

Then I suggest a paragraph for RacGTPases structure and function and sub-paragraph to describe each member of the Rac family. Again, I think that as is, the manuscript is difficult to follow and the reader is easily lost. Too many details about mutations of Rac members should be avoided because nothing add to the scope of the review.

References should be placed before punctuation. Some reference numbers are not in square brackets.

Usually abstract does not contain references.

Author Response

We thank the editors and reviewers for their time and thorough evaluation of our manuscript. We revised the manuscript based on the outstanding feedback and we believe the work is much stronger consequent to the helpful comments that were provided.

Reviewer 2 Report

The authors concisely described the molecular characteristics and functional roles of Rac family, focusing on the mechanisms of atherosclerotic calcification and subsequent therapeutic strategy. This review manuscript is well written and comprehensive. This reviewer would recommend the authors to address a couple of points as shown below.

#1  What is the mechanism that IL-1β promotes vascular smooth muscle cells (VSMCs) differentiation into osteoblasts-like cells?

#2  Is it possible for Rac in VSMCs, but not macrophages, to be involved in VSMCs differentiation?

Minor:

Line 240, 298, 308, 554, and 579, please correct with square brackets.

Author Response

(The authors gave the same response as above.)
